# Community Involvement and Compensation Money Utilization in Ethiopia: Case Studies from Bahir Dar and Debre Markos Peri-Urban Areas

**Sayeh Kassaw Agegnehu [1,*] and Reinfried Mansberger [2]** 

1   Institute of Land Administration, Debre Markos University, Debre Markos 269, Ethiopia
2   Institute of Geomatics, University of Natural Resources and Life Sciences, A-1190 Vienna, Austria;
    mansberger@boku.ac.at
*   Correspondence: sayehalem@gmail.com; Tel.: +251-987990455 or +251-911076654

**Abstract:** In this study the involvement of the community during expropriation and the utilization of the compensation money of the expropriated farmers are investigated taking Bahir Dar and Debre Markos peri-urban areas as case studies. Survey research methods were applied for data collection. The data were analyzed by means of descriptive statistics. According to the results, there is high land tenure transformation in both study areas. Even though the majority of the expropriated farmers got compensation payments, most farmers did not use their compensation money to found alternative income generating businesses. Just payment of compensation shall not be an end by itself. Technical and administrative supports for farmers are essential for the proper utilization of the compensation money. Communities affected by expropriation should participate effectively in the processes of expropriation and compensation in order to reduce the externalities of the process. For this to happen, the public authorities should prepare open public consultation meetings prior to expropriation and must exercise smart democracy during the whole period of the process.

**Keywords:** compensation; Ethiopia; expropriation; participation; support

## 1. Introduction

Although Ethiopia is considered as one of the least urbanized countries even from Sub-Saharan Africa [1] due to the majority of its population being rural dwellers, currently there is a high rate of urban expansion [2]. This high rate of urban expansion caused high land tenure transformation and land use dynamics. Land, especially in the peri-urban areas, is subjected to high expropriation for different land uses including residential dwellings. The urban administration is expropriating peri-urban agricultural land to transform it to urban land up on payment of compensation.

When land is expropriated for urban expansion, participatory land expropriation practices and adequate compensation payments are essential. Community participation in land expropriation and compensation builds trust and reduces land disputes [3]. Accordingly, communities affected by expropriation should effectively participate in the processes of expropriation and compensation [4]. It is explicitly stated even in Proclamation 455/2005 that it is mandatory to inform the landholders with a written letter about the time of expropriation and the amount of compensation payment at least three months in advance.

The basic principle of payment of adequate compensation is to keep the landholders in the same economic position after the land is expropriated [5,6]. Landholders should get adequate compensation for the land taken. In this way, they would not be hurt as well as not gaining extra benefits at the expense of others [4,6]. It is a legal requirement of many countries that fair and reasonable compensation has to

be paid to parties affected by expropriation. This is also beneficial for the sustained development of a nation [7]. However, in most East African countries it is observed that expropriation procedures are not transparent, compensation payments are either not paid or are inadequate, and the participation of the affected property holders is missing [8]. As observed by Alemu [9], in the current situation when the majority of the population is living in rural areas, urbanization in Ethiopia is a "not only necessary but also unavoidable activity". Thus, the expansion of the urban will continue. Urban administrative bodies are expropriating peri-urban land and the displacement of the peri-urban farmers is increasing. Expropriation may be beneficial to society but it disrupts those who surrendered land, and the problem becomes worse if a sound expropriation and compensation system is not in use. In the absence of adequate compensation payment, expropriation leads to landlessness, loss of livelihoods and increased poverty of the expropriated farmers [10]. Inadequate compensation payments [9,11,12] and a lack of delivering the necessary support on compensation money utilization [13] are suggested as critical problems of expropriation processes in Ethiopia.

Sustainable development can be achieved if there is a harmonious interaction of economic, social and environmental priorities between urban, rural and peri-urban areas. Peri-urban as a space is an interface between urban and rural areas. The spatial expansion of cities is dependent on the expense of peri-urban landholdings. When peri-urban land is expropriated for cities expansion, agricultural households, especially those in developing countries whose livelihood is dependent on the fragmented landholdings, are exposed to the reduction of their farm sizes and, as a consequence, the diminishing of production. This leads to poverty of the peri-urban farming community. Besides, the social wellbeing will be disrupted and good governance can be challenged if the expropriation and compensation procedures are not properly implemented. Sound and smart expropriation and compensation rules are required to improve or at least to maintain the quality of life of the affected households. Accordingly, the expropriation system, the practices of expropriation and compensation as well as the support delivered to expropriated farmers have to be studied and corrective measures must be proposed to enable the expropriated farmers to be economically sustained. Compensation payment for displaced landholders must enable those who surrendered land to gain equitable benefits that they lost from the land [4,14,15]. As depicted above, the majority of the Ethiopian people are peasants and agriculture is the basic asset of the economy. Therefore, if the issues concerning land expropriation and compensation are not properly addressed, serious socio-economic problems will arise affecting the livelihood of peri-urban farmers and the stability of the nation.

The displaced property holder shall be properly compensated for the losses they suffered due to expropriation. Therefore, the designing of fair and appropriate expropriation systems is of paramount importance. The focus of this study is to identify the main shortcomings associated with the execution and legislation of the current compensation. Specifically, we (i) assessed the perception of land administration experts and victim farmers on the amount of compensation, (ii) analyzed the support delivered to affected households in using the compensation money for income generating businesses, (iii) assessed what the community involvement in the expropriation process looks like, (iv) investigated how fairly the current expropriation and compensation payment is executed, and (v) checked local impacts by carrying out the study in two test sites. The article also aims to give evidence about divergences in the expropriation process in theory and practice. The awareness of the documented differences helps to adjust the current processes to fulfil the legal framework. In addition, the findings of this research can be used as an input for policy makers in designing expropriation and compensation legislation fit to purpose, especially in Debre Markos where this kind of investigation was done for the first time.

## 2. Literature Review

### 2.1. Expropriation

Property rights can be owned privately or possessed by individuals with usufruct rights depending on the legal context of the country. Ownership rights allow the right of alienation on the property, whereas possession rights cannot permit alienation of the property right and are considered as ill-defined rights. Even if significant differences can be observed between these two rights, property rights are not absolute since governments in different countries expropriate land for public purposes [14]. That is, the state has overriding interest in property rights in order to construct infrastructures and public facilities, to make necessary adjustments on economic and social inefficiencies in private and market operations and to balance the power difference between poor and elite groups in resource utilization [10].

The process of taking private property for public purposes has different connotations in different countries. For instance, the term expropriation is most commonly used in Continental Europe, whereas other used terms include eminent domain (United States), compulsory purchase (United Kingdom), resumption/compulsory acquisition (Australia) and/or resumption (Hong Kong). Ethiopia uses the term expropriation and, accordingly, in this study, the term expropriation is used preferentially.

Expropriation is the power of taking private property rights mostly by the government without the consent of the owner, paying compensation when that land is required for public purposes [4]. This power is often necessary for the socio-economic development and environmental protection of a country. Nevertheless, it should be balanced between the public and private interests since it adversely affects land tenure security if not properly implemented [4,16,17]. Expropriation is a proper tool for most countries when they require land for public purposes. Though land can also be arranged by other means, such as voluntary agreements [6], the government cannot rely only on voluntary transactions for the timely implementation of infrastructures and facilities, which are important for the socio-economic development of a country [4]. While expropriation was widely used traditionally for public infrastructures and facilities, such as public roads, parks, schools and health care [16,17], currently in Ethiopia, expropriation is widely applied when land is required for urban expansion. Governments undertake expropriation when infrastructures, utilities and public facilities are planned to be built, both in rural and urban areas.

In Ethiopia, the main reason for expropriation of land is the fast-rate horizontal expansion of cities [2,18]. In order to provide land for residential, commercial, industrial buildings and associated infrastructures and facilities, vast tracts of peri-urban land are expropriated annually. Residents form housing associations and ask the municipality to give them land for the construction of residential houses. After some years, the municipalities expropriate land from peri-urban landholders and transfer it to the housing associations. The municipalities are delivering land for the development of villa residential houses. There is no system which guides the residents to practice condominium buildings in associations. Even the inner city is not well-developed. What is being exercised is to expropriate peri-urban agricultural land and to transform it into urban land use types. Even though the general municipal-level scenario looks like this when expropriating land for public purposes, adequate compensation should be paid [6]. This has been enacted at international level as a human rights issue.

At international levels, the payment of compensation during expropriation of land for public purposes has been addressed, not only as an economic but also as a human rights issue since property rights determine the livelihood of the people. Article 17 of the International Human Rights Law [19] highlights the principle that "everyone has the right to own property alone as well as in association with others" and that "no one should be arbitrarily deprived of his property". "Land rights as a human right are described as components of the right to an adequate standard of living, which entails the right to adequate housing, and the right to adequate food" [20]. Therefore, it is an international covenant to provide "effective remedy" for the evicted property owners during expropriation [21]. In implementing this international law, different countries have taken various measures. On the European level,

the European Convention on Human Rights [22] makes a statement about the protection of private possessions and dictates that if depriving of property in the public interest is required, this has to be according to the law and the guiding principles of international law. The protection of private property, provision of a fair trial, and protection of a right with respect to a house are the three fundamental issues addressed in the European Convention on Human Rights with regard to property.

Developing countries, for most of which land is the basic asset of the economy and the basic means of livelihood for the society, are also expected to use this international law as a guiding principle and to abide to the international laws. For different reasons, most countries have different laws to govern their land. For instance, East African countries have different legal systems to govern land depending on their historical, cultural religious norms [8]. IGAD (Intergovernmental Authority on Development) has stated that expropriation of land for public purposes shall be based on law and on payment of compensation.

## 2.2. Valuation for Compensation

In order to guarantee adequate compensation, the designing of sound valuation system is very essential. The system principally can be based on market value under current use or on expected value increases due to change of the current land use [23,24]. The most commonly used payment of compensation is based upon current market value [3]. Market value is the estimated amount on the date of valuation for which an asset should be exchanged between a willing buyer and a willing seller in an arm's length transaction after proper marketing, wherein the parties have each acted knowingly, prudently and without compulsion [4,25]. The three conventional approaches to the estimation of the market value of real properties are the sales comparison approach, the cost approach, and the income approach [25]. The sales comparison approach is a method to estimate the value of the subject property with the price of similar and recently sold properties. This system of valuation can be applicable if sufficient data on recent market transactions are available. The cost approach of valuation is a valuation system based on the replacement cost of the property. It assumes that the market value of a new building is similar to the cost of constructing it today. The income approach can be used to value commercial and agricultural properties. It also can be applied as a substitute of comparable sales approach in situations of low market transactions.

## 2.3. The Legal Basis of Land Expropriation and Compensation in Ethiopia

In Ethiopia, the legal basis for expropriation of land is enacted in the 1960 Ethiopian Civil Code. For instance, in article 1460 of the civil code, it is stated that the expropriation proceedings can be applied if land is required for public purposes. Again, the compensation payments for real property are enacted mainly in the articles 1470 to 1479. As stated in these articles of the civil code, not only personal but also real property compensation is legally permitted. The compensation can be either payment of money or another land. Nevertheless, Article 1474 of the Code states that in any case compensation has to be equal in value to what is expropriated or to the actual damage occurred. However, after the overthrow of the Emperor Haile Selassie by the military junta (Derg) in 1974, the situation changed. Between 1974 and 1991, the Derg regime followed socialist doctrine and handled expropriation by enacting an own land reform proclamation [26]. In this proclamation, the rules about expropriation and compensation in the civil code are not followed.

The power of expropriation was vested to the government as clearly stated in proclamation 31/1975. In Article 17:1, it is stated that "the government may use land belonging to peasant associations for public purposes such as schools, hospitals, roads, offices, military bases and agricultural projects". In other words, the state was the sole expropriator of land without payment of any compensation. Number 2 of this article states that the damages that might be caused by this expropriation decision on the peasant association are considered to be taken for good.

Currently, the basis for expropriation and compensation of the property rights issue is set in Article 40 of the federal constitution of Ethiopia. In the constitution, it is stated that all urban and rural lands

as well as natural resources are owned by the government and people of Ethiopia with no time limit for peasants' landholding, i.e., farmers have usufruct rights in perpetuity. With this article, the government is given power in order to expropriate private property upon payment of compensation.

The expropriation, as well as valuation and compensation systems, is set in proclamation 455/2005 [27] and in its subsequent regulation [28]. The power of execution of the proclamation has been decentralized to the lower administrative bodies. Wereda (wereda is a lower administrative structure in Ethiopia, just like a district) rural and urban administration offices can undertake expropriation when a need for land arises on payment of compensation. When land is required for urban expansion, the urban administration office makes decisions and the urban municipality facilitates activities for expropriation. This procedure is similar for rural areas with the rural administration and land administration offices as counterparts.

Expropriation should be handled with a legislation with procedural rights of the affected people including right of notice, the right to be heard and appeal, in order to reduce its enormous economic, social and political costs [4]. In proclamation 455/2005 it is stated that property holders should participate actively during expropriation.

As stated in Article 7:1 of proclamation 455/2005, a landholder can receive compensation for the "properties situated on the land and permanent improvements he made on that land". The method of valuation for the properties situated on the land is based on replacement cost of the property. The valuation for permanent improvements takes into account the value of labor and capital spent to establish these improvements. If the property is to be erected in another area to continue its service as before, compensation must be paid for the removal, transport and relocation of the property (Article 7:5 of 455/2005). The expropriated landholder permanently losing his land also receives "displacement compensation", which is ten times of the previous five-year average annual income as described in Article 8 of 455/2005. The five-year average income is calculated by the annually timely market prices and multiplied by the factor 10. This amount of money is paid as compensation to the expropriated property holders. Certified private or public institutions are permitted to conduct the valuation according to proclamation 455/2005. But due to lack of availability of these institutions, the current valuation process is being carried out by committees and by the owners of utility lines. In proclamation 455/2005, it is also explicitly explained that delivering rehabilitation support is one of the main responsibilities of the wereda administration. Accordingly, the wereda administration should provide technical and administrative support to mitigate the livelihood disruption of the affected households.

## 3. Study Areas and Methodology

### 3.1. Study Areas

The research is conducted in the peri-urban areas of the cities of Bahir Dar and Debre Markos (Figure 1). The data for the research are collected mainly in the peri-urban areas of these cities within a radius of 5 km from the periphery of the dense urban areas.

These two towns were selected in an effort to observe differences in the relative rates of spatial urban expansion and its effects between a big-size city (Bahir Dar) and a medium-size city (Debre Markos) in Ethiopia. Even though, as practically observed and reviewed from literatures, the urban areas in the region and also in the country are expanding at a relatively high rate, the rate of expansion and land expropriation seems high in big-sized and medium-sized cities. Thus, the main reasons for the selection of these study areas are the changes in land tenure transformation due to rapid urbanization, the high demand for land, the fair representation relative to other big-sized and medium-sized cities in the region, and the ease of collection of the necessary data within limited time and financial resources.

A heterogeneous mosaic of agricultural, forestry, meadow, and residential land use types dominate the land-use pattern of the study areas. Nevertheless, each study area has its own local characteristics.

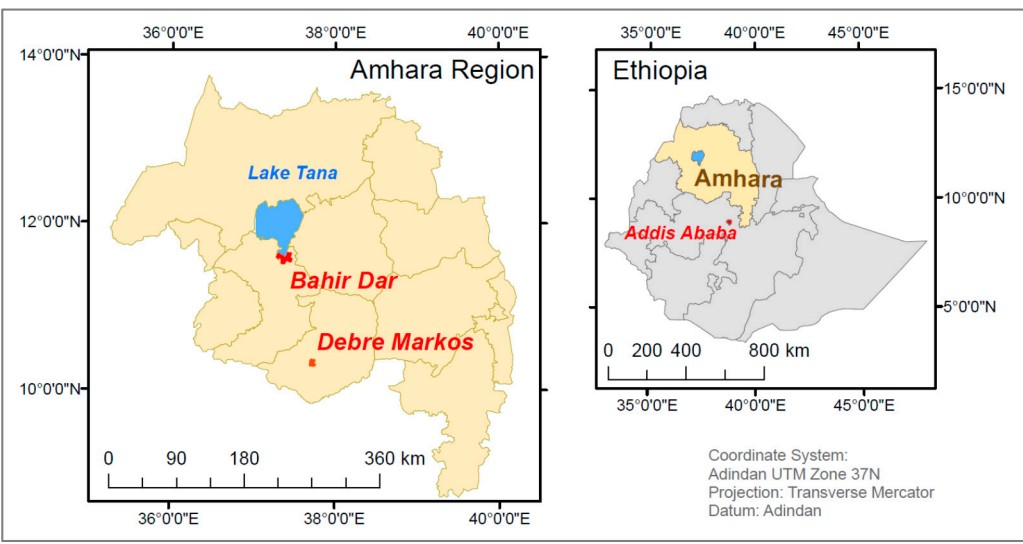

**Figure 1.** Map of the study areas.

Bahir Dar (located 11°36′ North 37°23′ East, with an average elevation of 1800 masl) is the capital city of the Amhara National Regional State (ANRS) of Ethiopia. It is located to the southern shore of Lake Tana in the western part of ANRS, and in the northwestern part of Ethiopia. The name Bahir Dar, which translates to 'Sea Shore', is derived from the proximity of this city to the lake. Topographically, Bahir Dar is located more or less in a flat terrain, although there are some slopes to the western and eastern peripheries. The historical foundation of the city was inherently linked with the establishment of the church Kidane Mihret in the 14th century [29]. The modern development of the city began in the 1930s after the Italian occupation. The city was used as a military base during the occupation. As a result, the construction of an airport, army camps, administrative offices, residential housing and business centers was launched. After the Italian force was defeated, Bahir Dar became the place of the wereda administration, then the capital of the Awraja (administrative organization between regional state and wereda), and finally the capital city of the ANRS of Ethiopia.

Debre Markos is located in the southern part of ANRS and 300 km in the northwestern direction from Addis Ababa, at 10°20′ North and 37°43′ East, with an average elevation of 2400 masl. Debre Markos is the capital of the East Gojam administrative zone. Topographically, Debre Markos has a mixed scenario of flat and sloppy terrains. The historical foundation of Debre Markos was in 1852 by Dejazmach Tedla Gualu, who was administrator at that time. The initial name of Debre Markos was Menkorer. King Tekle Haimanot changed this name to Debre Markos in 1869 when the principal and historical church was constructed and dedicated to St. Mark. Debre Markos was the capital of Gojam province during imperial times and for most of the Derg regime.

*3.2. Methods of Data Collection*

The data presented in this study comprises the period during which ANRS started to administer rural land by enacting proclamation 46/2000 up to 2014. Survey data collection is employed for this research. Survey is the most widely used data gathering technique and it is applied to measure many variables, to test hypotheses, and to infer temporal order about past behavior, experience, or characteristics [30]. The techniques of survey data collection can be face-to-face interviews, telephone interviews, mail questionnaire, self-administered questionnaire, and/or web surveys [30]. In this study, face-to-face interview and self-administered questionnaire are used to collect data from farmers and experts, respectively. In order to select the study households, initially, the total household list of the study peri-urban areas was obtained and recorded from the study sites concerned land administration offices and agricultural development agents' offices. From the list, 2386 households that lost land through expropriation are identified and recorded with the support of kebele (smallest administrative

unit) land administration committees. Respondents were selected randomly from the expropriated household lists in both study areas. In total, 269 respondents were selected and interviewed with the aim to assess the expropriation and compensation practices. Out of these respondents, 101 are from Debre Markos peri-urban areas whereas the remaining 168 are from Bahir Dar peri-urban areas.

In addition, questionnaires were used to assess the opinions of land administration experts about the current compensation payments and the utilization of the compensation money. Most of the experts were engaged in valuation and compensation committees during expropriation processes. The experiences of these experts are useful in order to formulate recommendations which will improve the problems associated with compensation payments and compensation money utilization. The survey questions were delivered to 26 professionals. Out of those, three did not return the questionnaire in time. All survey employees are from municipalities and newly established rural land administration processes in the urban administration of both study areas. All experts have exposure to the expropriation and compensation practices and rules with different years of experience in the institution, the minimum being six months and the maximum seven years. The survey employee list comprises both experts and coordinators (vice heads, process owners) at various levels. About 22% of them are coordinators of land administration processes at different levels. Briefly, the survey group represents professionals at the management and technical levels with various levels of experience.

## 4. Results

### 4.1. Peri-Urban Arable Land Loss

As observed in the study, there is a high rate of land transformation from peri-urban agricultural landholdings to urban built-up areas [2]. As a consequence of this, many households have lost land and the landholding has been substantially reduced. The sampled 269 respondents have lost 441 parcels of land of about 260 ha due to expropriation as a consequence of urban expansion. On average, every respondent whose land is expropriated has lost about 0.95 ha of land, and the average landholding of these farmers has been reduced nearly by half. The current average landholding of the respondents is now about 0.97 ha. The pressure that the expansion of the urban posed is not only a reduction of landholdings in peri-urban areas; there are also farmers who lost all of their landholdings. About 3% of the respondents replied that they had lost almost all of their landholding because of urban expansion.

### 4.2. Pre-Informing of the Community and Timely Compensation Payments

Pre-information about the intended expropriation as well as notification with written letter to expropriated households were poorly implemented in the study areas. Out of the total respondents, only 15% replied that they were pre-informed about the expropriation of their land (Table 1). When the two sample areas are analyzed, the situation in the regional state (Bahir Dar) is better than Debre Markos. It is possible that the frustration due to the relatively high rate expansion of Bahir Dar city might have enabled the peri-urban poor to dig and get information about the expropriation. Otherwise, if prudently done, the knowledge of households about expropriation and compensation rule (Table 1) would not be lower than in Debre Markos.

**Table 1.** Community participation.

| Data Types | YES Responses (in %) | | |
| --- | --- | --- | --- |
| | **Debre Markos** | **Bahir Dar** | **Total** |
| Pre-information for expropriation. | 10.0 | 17.3 | 14.6 |
| Do you have some knowledge about valuation and compensation rules? | 23.7 | 18.5 | 20.2 |
| Have you got the compensation money in time? | 75.2 | 84.1 | 80.8 |

The bases are all respondents (101 in Debre Markos, 168 in Bahir Dar, 269 in total).

The majority of the expropriated respondents (80%) did not have at least a general knowledge about the expropriation, valuation, and compensation rule. Even the remaining 20% had knowledge only to some extent. The study tried to assess why they did not have knowledge on these processes. The majority of the respondents said that nobody told them about the expropriation and compensation rule and that they were not invited for a meeting to discuss on expropriation and compensation. Other farmers said that it may probably be the lack of their presence in the meetings because of being busy with other duties. Study respondents were also interviewed about the timely payment of the compensation. About 80% of the respondents replied that the payment of the compensation was made in a timely manner (Table 1).

*4.3. Opinion on the Amount of Compensation Payments*

As seen from the above result, even though the response of the respondents on the timely payment of compensation is positive, most of them were highly dissatisfied about the amount of the current monetary compensation payment. According to the ranked assessment result of their perception, about 88% of the respondents reported that the amount of the compensation payment is low, as shown in Table 2. Only about 3% of the respondents ranked it as good while there was not a household who rated it as very good.

**Table 2.** Respondents' perception of the current compensation payment.

| Scoring | Percent | | |
|---|---|---|---|
| | Debre Markos | Bahir Dar | Total |
| Good | 6.0 | 1.8 | 3.4 |
| Fair | 7.9 | 8.3 | 8.1 |
| Low | 85.1 | 89.9 | 88.1 |
| No Answer | 1.0 | - | 0.4 |
| Total | 100.0 | 100.0 | 100.0 |

The bases are all respondents (101 in Debre Markos, 168 in Bahir Dar, 269 in total).

Even though the ranked assessment on the amount of compensation money is valued negatively by both sample areas, when we see the data from the two sample respondents, by comparison, it is better in Debre Markos. This can be related with the lease price of land. In Bahir Dar, the lease price of land is more expensive than in Debre Markos.

*4.4. Compensation Payment*

In the study, the compensation payment situation was investigated. About 91% of the respondents received full displacement compensation during expropriation (Table 3). The remaining 9% constitute respondents who lost their land without compensation and/or received a very small amount of compensation compared to what is stipulated in the proclamation. In other words, some respondents received displacement compensation not for ten years, but for one year only.

**Table 3.** Compensation payment.

| Data Types | Percent | | |
|---|---|---|---|
| | Debre Markos | Bahir Dar | Total |
| Have you got full compensation for expropriated parcel? | 74.3 | 100.0 | 90.8 |
| Type of Compensation | | | |
| Money | 74.7 | 94.8 | 88.7 |
| Money and urban residential land | 25.3 | 5.1 | 11.2 |

The bases are all respondents (101 in Debre Markos, 168 in Bahir Dar, 269 in total).

The two main types of compensation identified in this study are monetary compensation and urban land compensation. However, the main compensation type was monetary compensation. Only 11% of the respondents received urban residential land and some amount of compensation money. In both of the study areas, some households were compensated with both urban residential land and monetary compensation.

### 4.5. Use of the Compensation Money

In this study, the support given to affected households on how to use the compensation money was assessed. According to the study result, out of the 243 respondents who received full compensation, the majority (89%) did not get any support for using the money on alternative income generating businesses. The rest of the respondents (11%) reported that the kind of support was simple information on using the money properly (Table 4). Otherwise, none of the respondent replied that they got basic support in preparing alternative feasible projects, or delivering some training, or some advice for participating in projects.

**Table 4.** Compensation money utilization.

| Data Types | Percent | | |
| --- | --- | --- | --- |
| | Debre Markos | Bahir Dar | Total |
| Money Utilization (multiple responses) | | | |
| ■ Regular expenses | 80.0 | 66.5 | 70.6 |
| ■ Income by generating businesses | 17.3 | 23.7 | 21.6 |
| ■ Saving in bank | 21.3 | 26.6 | 25.3 |
| Support for money utilization | 9.3 | 11.0 | 10.5 |

The bases are respondents who received compensation (75 in Debre Markos, 168 in Bahir Dar, 243 in total).

Accordingly, the received amount of money often is not used properly. Most of them have used the compensation money for regular expenses (71%); some of them have kept it in the bank; whereas a few of the respondents have used it for income generating businesses (Table 4).

### 4.6. Opinion of Land Administration Experts on the Amount of Compensation Payment

A survey about the opinion of land administration experts on the current compensation payments was conducted. This was done to understand their perception on the current expropriation and compensation legislation; to assess their views on the rehabilitation support to affected households; and to understand their opinion on the impact on land tenure security. The opinion of the employees about the amount of monetary compensation payments was judged using ranked scales. From the received answers (21 employees in total), there is no employee who judged the current compensation payment as very good, but some employees reported that the current compensation payment is good. About 57% of the employees reported that the current compensation payment is not enough. Almost all the experts gave evidence that there was no administrative and technical support for the affected households to use the money for income generating businesses. In addition, these experts were also asked their opinions about the impact of the urban expansion on peri-urban landholdings. The majority of these employees (83%) reported that the expansion of the urban is creating land tenure insecurity threats on the landholdings for peri-urban farmers.

## 5. Discussion

### 5.1. Land Expropriation and Peri-Urban Land Transformation

In Ethiopia, expropriation is the only means of land acquisition for public purposes as purchase of land is legally prohibited. Therefore, the government is the sole supplier of land needed for public purposes. The current main segment of expropriation in the study of peri-urban areas is in order to

provide land required for urban expansion. Up on the expropriation of land, municipalities transfer the land to different parties for development. This can be done either by administrative allocation or through lease agreements. Previously, land for different purposes was transferred to acquiring persons freely by administrative decisions. Currently, the urban land is not allowed to be held without lease and the lease price can be determined by tender auction [31]. Land price is determined based on a competitive basis with tender. So, land transfer by lease agreements is principally based on market value. However, as being observed in the competition, usually the high lease price is not affordable for the low-income urban population. This competition is catering to rich people and even these people are making the competition fierce. Consequently, the government is designing a system which enables the poor to get land for housing by administrative decisions. Whatever the transfer might be, land in peri-urban areas is under a high rate of transformation from agricultural to urban land use types [2].

The reduction of the landholding in peri-urban areas as a consequence of expropriation due to urban expansion is also a challenge for continental Europe. For instance, Lennert et al. [32] described a high loss of peri-urban agricultural land in Budapest because of urban sprawl. The authors stated that this is a challenge not only for Hungary but also for Continental Europe, which the Common Agricultural Policy (CAP) reform should give due attention. With respect to Ethiopia, the case has been noted by other similar studies conducted in the country. For instance, Haregeweyn et al. [29] noted that from their Bahir Dar peri-urban study households, on average, every respondent has lost 0.89 hectares of land. Other studies conducted in the peri-urban areas of Addis Ababa city also documented that the landholdings of peri-urban farmers have been substantially reduced due to expropriation as a consequence of the fast rate of urban expansion [13,33–35]. For instance, Nigusie [35] gave evidence that each respondent had lost on average 1.29 hectare of land due to expropriation conducted between 2006 and 2010 in the Sebeta peri-urban area of Addis Ababa. According to his study result, about half of the respondents have hold of and work on less than 1 hectare of agricultural land. Kasa et al. [13] also observed that the Addis Ababa city built-up area is expanding dramatically, with a characteristic of horizontal spatial expansion consuming many hectares of agricultural and forest land. Thus, the result of this study and the review of other previous studies indicate the existence of high land tenure transformation in peri-urban areas of Ethiopia due to the current fast rate of urban expansion. The trend also indicates that it is increasing from time to time, even though the extent differs based on the standard of the cities. In big cities like Bahir Dar and Addis Ababa, it is increasing at a high rate and in medium sized cities, such as Debre Markos, there is an optimal rate of increase.

In this study, the livelihoods of those respondents (3%), who lost all their agricultural land as a consequence of expropriation, depend on land rental transactions and daily labor work. Other scholars have also noted this loss of landholding due to expropriation for urban expansion. For instance, Tadele [33] assessed the impact of real estate projects in the peri-urban land tenure and the livelihood situation of dispossessed farmers in peri-urban Addis Ababa. According to his study, a single real estate project has displaced about 125 households, out of which about 22% households totally lost their land rights, including residential houses. Fransen [11] observed a similarly high rate of land transformation from agricultural land to urban land in Ethiopia because of urban expansion.

Ethiopia has given significant attention to strengthening micro- and small-scale manufacturing enterprises. This is documented in the current five-year growth and transformation plan, as well as in previous plans. Enterprises are promoted to create job opportunities and prosperity in the country. Because of the great desire to accomplish agriculture-led industrialization in Ethiopia, the government motivates internal as well as external investors to realize their project ideas of establishing industries. Accordingly, priority is given to delivering land for them. However, some of the investments are speculative, resulting in a transformation of agricultural land to unused land [11]. Therefore, the land can be left idle for years without being used in any way. Such kinds of parcels are found in Bahir Dar and Debre Markos, whereas the management and utilization of such land varies between both study areas.

In Bahir Dar, these areas are temporarily utilized. The municipality of Bahir Dar allowed the farmers to cultivate the land until the launch of the development. However, in Debre Markos peri-urban areas, these speculated parcels are kept bare. Most of the respondents in Debre Markos raised their grievances, saying that their productive agricultural land is expropriated but kept idle without launching any development on it for more than six years. They also reported that they asked the municipality to allow them to temporarily use the land until launch of development, but they were prohibited from cultivating. This issue was discussed during the field study with the Debre Markos municipality process owner. He argued that they cannot allow farmers to cultivate the land once the compensation is paid as the developers may not be patient to wait until the crop is harvested. This opinion is shared with some other experts too. Both the municipality experts and the managers are ambitious to attract investors with on-time delivery of land, putting aside the needs and aspirations of the peri-urban subsistence farmers. This is simply a lack of flexibility and a lack of decision-making competency to solve local problems according to the local context. Otherwise, from a practical point of view, those who need land for development might not be negligent in waiting until the crop is harvested which might be about three months. In order to reduce the skepticism of municipalities, it would be appropriate to make an agreement with farmers which enables them to cultivate the land until the development launches.

## 5.2. Community Participation in Expropriation and Compensation

It is necessary to pre-inform the households which are to be affected by expropriation that their parcels will be handed over for public purposes. This enables them to be mentally ready at the time of the process and to think about other alternative livelihood strategies in order to generate income that they may lose from their land. Besides, the satisfaction level of affected households will increase if they are invited and involved in discussions in advance in the expropriation meetings [36]. However, it is identified in the study that there was no participation of the affected people in expropriation and compensation processes. This lack of participatory expropriation and compensation payment practice has been also noted as one of the main problems of land administration in East African countries, which needs to be resolved by designing appropriate land policy [8]. Other similar studies conducted in Ethiopia also documented similar problems with respect to the involvement of the affected community [15,34,37,38]. The result of our study and other similar research findings in Ethiopia indicate that, even though participatory approaches are necessary in the expropriating of land for public purposes, the reality at the ground indicates that it is overlooked by the municipalities. The trend should indicate improvement but, as has been noted in the current studies too, significant improvement is not observed. Ambaye [39] observed that, in general, the Ethiopian expropriation proclamation by itself has shortcomings in clearly dictating the procedural steps to be followed during expropriation. However, at least from a rights perspective, it is explicitly stated in the proclamation that expropriated landholders must be informed with a written letter about the expropriation of their parcels. Involvement of the landholders to be expropriated is also beneficial to gain the necessary information to be considered during valuation. Recognizing farmers' "right to know, right of expression, right of supervision, and right of benefit" is also the basic requirement for the amicable execution of expropriation [40]. However, respondents described that there were few discussions with them during the expropriation process. Instead of the involvement of the affected community, local administrative representatives represented the society during expropriation and compensation processes. The availability of local administrative bodies is not a problem but they cannot substitute for the affected households. Thus, the valuation committees conducted the valuation of expropriated parcels and improvements involving local administrative bodies. Afterwards, the amount of compensation money to be paid was posted to the kebele office.

Based upon the information, the expropriated landholder will accept or refuse the proposed compensation money. In cases they refused, the money would be deposited in a block bank account by the name of the rural or urban administration office until the case is decided. If landholders refused to

take the money assuming that it is not properly valued, they can present their complaint to the grievance hearing administrative organ of the urban administration or to the court office in rural areas [27]. If still dissatisfied with the decision, they can appeal to the appeal court. If the size of the parcel is not properly considered and the crops value not properly estimated, the decision-making bodies have a right to make adjustments by assigning other experts who will scrutinize the case. Even if the compensation money to be obtained by complaint might be a small amount, affected people get a better feeling if their grievance is handled properly. Landholders are liable to handover the expropriated land within a maximum of three months after they receive the compensation. The three-month period is enforceable if the land is covered with crops or occupied with other properties. If the land is not occupied with crops or other properties, landholders have to relinquish the land within a month after the payment of compensation. As noted above, during expropriation nobody was informed officially with written letters.

Respondents, who were not pre-informed by the public administration, were asked when and how they knew about the expropriation of their parcel. The majority of the respondents replied that they became aware when the process of expropriation was launched, i.e., at the time of surveying and notification of the amount of compensation money. Some others had heard rumors about expropriation by informal means, but they were not sure whether the expropriation was to be carried out or not.

*5.3. Adequacy of Compensation Payments*

Of course, however generous and however precisely expropriation procedures are executed, landholders in general are not happy if their land is handed over [4]. For them, it is the land where they have lived for many years and which they expect to transfer to their heirs. However, the dissatisfaction becomes relatively high if the compensation to be paid is low [35]. The study respondents' perception about the reasons for dissatisfaction is divided into three categories. There are farmers who perceive that the problem is due to expropriation and compensation directive. There are also farmers who perceive that the problem is caused by imprecise measurement of their parcels during the expropriation process, while others see a problem in an improper inventory of the amount and type of crops produced by them. All the issues noted above have their own influences. The valuation system is not market-value-based; the surveying for estimating the size of the parcel is carried out simply by traditional means using measuring tapes, which does not enable to measure irregularly shaped parcels precisely; and the type of crops planted during the previous five years are not properly recorded since affected people were not consulted in time. Other similar studies conducted in peri-urban Bahir Dar observed similar results and have recommended that compensation payment should be based on the market value of lands [15,18,41].

In Ethiopia, land is not a free commodity subject to exchange through sales. Therefore, it is difficult to compare explicitly the price of land and the compensation payments. However, there are illegal land transactions in the peri-urban areas. In addition, currently the transfer of urban land for different uses is being done by means of leases. So, it is possible to make a rough comparison between the compensation payments with the price of the land in illegal transactions and lease prices to judge the compensation payments. From what is practically observed, the prices for illegal transactions and leases are very high. The farmers' dissatisfaction increases when they hear the high lease prices of the adjacent parcels. Accordingly, they hate expropriation to a high extent and try to get advantages, even by transacting their land illegally. Illegal transaction is not secure, especially for the buyer, since if the seller denies the transaction, the buyer loses their money. These illegal transactions are based on trust and religious taboos. Even under such uncertainties, the illegal sales price is by far better than the compensation payments and, accordingly, peri-urban farmers are motivated to undertake illegal sales before their land becomes expropriated.

It is not only the affected farmers who are requesting revision of the expropriation and compensation legislation. Most of the land administration experts interviewed in this research also replied that the amount of the compensation is low. To resolve the shortcomings, most of them proposed a revision

of the compensation payment legislation by suggesting the implementation of new legislation that estimates the value of the land properly. Some others envisioned that some modification on the current compensation payment directive is sufficient, suggesting that instead of considering the five-year average value during valuation, estimating the value of crops by taking the current market price is a better alternative.

During expropriation times, there are also false promises to the affected people by the municipalities. For instance, in Debre Markos peri-urban areas, many respondents complained especially that during expropriation, the urban municipalities promised to give priority to them for some advantages, such as job opportunities, electricity and pure drinking water supply. However, after the relinquishment of the land, nobody noticed the previous promises. Abdissa [34] also observed similar problems in his study. He noted that although the municipality promised the expropriated farmers different infrastructures and utilities, the promise was neglected after the land was handed over to the municipalities. Thus, avoiding false promises is important to build trust between the affected people and the expropriating bodies, as well as with other stakeholders who will contact the households for different duties.

### 5.4. State of the Compensation Payment

The failure of payment of compensation for the expropriated land has locational differences. As seen in Table 3 and another similar study [29], in Bahir Dar peri-urban areas all respondents whose land is expropriated received compensation. Bahir Dar is the capital of the ANRS and, accordingly, the peri-urban farmers have easy access to raise their issues up to the regional state if compensation is denied. Even the municipality was responsive to pay the compensation money. However, in peri-urban Debre Markos, some respondents did not get compensation, while some others paid a small amount. The basic reason for this is landholding claims between the municipality and the landholders in the urban fringe. The municipality of Debre Markos town considers some land in the urban fringe as its own holding which had been held by farmers for many years in the past [2].

As documented in Table 3, only a few respondents (11%) received urban residential land and some amount of compensation. The reason for this low rate can be explained with the fact that it was practiced only one time. Afterwards, the residential land compensation was omitted and only monetary compensation was paid. Most of those respondents who received some amount of compensation money and urban residential land have built residential houses and are getting some amount of money by renting houses and dormitories. Therefore, getting urban residential land as a supplement to the monetary compensation is a preference of these peri-urban farmers.

In this study, respondents were asked what kind of compensation system they deem appropriate for future expropriation. Concerning the monetary compensation, nearly all of the households required a revision of the current compensation legislation, complaining about the inadequacy of the current compensation payment. Nevertheless, land-to-land compensation is the primary preference of many respondents (67%). During expropriation, when the alternative for compensation is between monetary compensation and land-to-land compensation most often affected people require land-to-land compensation than compensation to money [4]. When vacant land is available in their village, the land-to-land compensation is a proper system of compensation even for the government. Nevertheless, there is usually no extra land available for land-to-land compensation in their village. Therefore, it was interesting to get knowledge of where the farmers think land might be available in order to undertake land-to-land compensation. The solution forwarded by some farmers during the interview was impressive. In general, they have noble solutions for local problems if they would be consulted in decision making. Some farmers aspire to launch mechanized farming practices in the nearby lowland investment areas provided that they get initial support from the government. This is really an incredible idea since the impact of their investment is multifaceted and would lead to more than their livelihood improvement.

However, the preferences of old-aged and young and/or middle-aged farmers differ. Most of the old farmers, especially those above the age of 60, suggested a better compensation payment if they do

not get land in their locality. The young and middle-aged farmers require the government to enable them to organize themselves in groups and to support them, and to give them land delineated and being used for investment in the nearby lowland areas. They are interested in getting credits to invest in the land and to start mechanized agricultural farming while they are still living in their residential sites. They want to get land in the nearby lowland areas and cultivate it in mobility.

*5.5. Support on Compensation Money Utilization and Use of Compensation Money*

Concerning compensation payment, the expropriation and compensation regulation states that the purpose is "not only payment of compensation, but also to assist displaced persons to restore their livelihood" [28]. However, concrete measures are not taken to achieve this purpose.

As noted in the result, the expropriated farmers spent a considerable amount of compensation money for their daily consumptions. Respondents described their previous intentions for how to use the money before the compensation was paid. Some were interested in being engaged in fattening and dairy farms, some were interested in being engaged in establishing grinding mills, and some were interested in getting sites for containers in the city to undertake different retailing businesses. Some were interested in renting land and undertaking agricultural activities as accustomed, while the rest was not clear about what to do with the money. Despite a lack of support, some of the respondents who received compensation payments are engaged in income generating activities. Utilizing the money for income generating businesses is a basic solution, which will enable the property holders to regain income lost from the land, since, as redundantly described, the basic reason for paying compensation is to enable the expropriated farmer to gain income commensurate with that lost from that parcel [24].

The income generating businesses are off-farm activities in urban areas or being engaged in other farm-related businesses in the village of the expropriated farmers. Both require strong support in designing alternative projects and developing the skills of farmers since these farmers are accustomed to a rural way of life and its customs. They are not able to utilize the compensation money profitably unless they get additional support. Most of those who used the compensation money for income generating businesses, did it by their own efforts. Some of them did not get some basic services when they required the investment of that money for income generating businesses. For instance, some farmers were unable to get electric power when they intended to install grinding mills.

This lack of the necessary support with compensation money utilization was also described by land administration experts. Most employees confirmed that the compensation money was misutilized by the farmers. They reported that the farmers were requested to take the compensation money without giving and even without thinking about any support for the utilization of the money.

These results support Keith et al. [4] who found that farmers affected by expropriation will use the money quickly and unwisely if the compensation payment is not associated with the necessary training on how to manage the money. Studies conducted in peri-urban areas of Addis Ababa also give evidence that expropriated farmers were not advised on how to use compensation money, neither by the governmental authorities nor by NGOs. Other authors [13,34,35] also noted that expropriated farmers did not utilize the compensation money in a livelihood improving manner because of lack of "parallel business and skill development interventions".

## 6. Conclusions and Recommendation

Countries have their own system in order to acquire land for public purposes. Some settle acquisitions by negotiation between the acquiring party and the landowner, while others settle them by official decision of the acquiring party. Whatever the system might be, adequate compensation payment to affected farmers is required and essential, at least to sustain the existing economic situation of the expropriated farmers in particular, as well as for the socio-economic development of the nation in general. The valuation system for compensation should be orientated according to the market-based values of land, and the compensation payments should aim to compensate the impacts of expropriation on the livelihood of the peri-urban poor. For the sake of this, continually revising and updating the

expropriation and compensation legislation is essential. Though expropriation for public purposes also takes place in other areas, special attention should be given in peri-urban areas where land is under particular pressure.

The high rate of land transformation in peri-urban areas is anticipated to increase steadily, especially in the least urbanized countries like Ethiopia. First and foremost, land has to be expropriated at the time when it is actually needed for development. In situations in which expropriated land cannot be developed immediately for different reasons, quick administrative decisions have to be taken regarding the temporary use of land by farmers for growing crops. Otherwise, leaving the productive agricultural land idle is not economical and hampers the farmers' ability to get some income from the land at least until the development time.

In general, land administration systems work well when the affected community participates actively. When expropriation and compensation are treated in a particular case, there should be public discussion meetings before the commencement of expropriation. This helps the government to design appropriate strategies to mitigate the negative effects of the process and to develop smart peri-urban villages.

For instance, when expropriation practices become participatory, the community builds trust in the government and can understand why the expropriation is being conducted; the governmental bodies can understand the needs and aspirations of the affected community which will be used as inputs to deliver the necessary support for the proper utilization of the compensation money. Pre-informing and participating of the community also enable the affected farmers to become mentally ready at the time of expropriation since displacement can not only disrupt the economic situation of the affected farmers, but also it has a great moral impact. It is also essential to improve the livelihood of the peri-urban farmers by taking different measures that improve the peri-urban village environment.

The basic principle of compensation is to enable the affected households to be kept in the same economic position that they were in before the expropriation. For this to happen, adequate compensation payment has to be paid. The valuation system for compensation becomes fair and efficient if the market value of the property [42] is used. Thus, valuation for compensation must be based on market value of the property expropriated. In order to undertake the market-based valuation, lack of availability of land sales might be considered as a pitfall. As, currently, urban land is being transferred by means of lease auctions, lease prices can be used to estimate the market value of the expropriated land. In addition, as seen from the analysis, the current compensation payment is not even properly implemented in some peri-urban areas at the lower administrative level. Thus, monitoring and control of the proper implementation of the designed system are necessary.

The current compensation system for peri-urban areas is often based on compensation with money. Besides, land-to-land compensation also has to be taken into consideration. This has been put forward as an alternative in the expropriation and compensation legislation if land is available. However, it is difficult to get land in peri-urban areas for land-to-land compensation since there is no unoccupied land in these areas. Nevertheless, compensation in land in other nearby unoccupied areas has to be taken as an alternative based on the interest of farmers who lost land through expropriation. For instance, in the lowlands of Ethiopia, there is vacant land which is delineated to be used for agricultural investment. Organizing voluntary farmers and delivering land to these vacant areas has also to be considered as an alternative. If young farmers intend to undertake diversified agriculture in the lowlands, the government should provide some loans for agricultural machinery, basic infrastructure, and some basic training to enable increased agricultural productivity. This has an immense economic value both for the affected households and for the nation. In addition, creating employment opportunities will be beneficial for the young farmers in particular, and for the country in general. The main constraining problem here is the financial issue, as these farmers will need money to purchase machinery. Supporting those farmers with credits, especially during the establishment phase of the investment, could resolve the financial problem.

Payment of compensation shall not be an end in itself. The proper utilization of the compensation money has to be given more attention. As described in the analysis, most of the respondents were not clear on how to use the compensation money, and were not capable of doing so. Because of this, most of them spent it for regular expenses. It would be beneficial if households would use the money in other income generating businesses. The affected farmers should get the necessary support with this, at least to commensurate the income they lost from the expropriated land. Technical and administrative support is required for the expropriated farmers, to show them how to use the compensation money in a proper and sustainable way and to improve or at least maintain their livelihood. Expert groups of the authorities have to design training and consultation for the expropriated farmers. In addition, administrative support is required to meet the challenges associated with the implementing and vitalizing of income generating activities.

This research has some limitations. The data presented is from big and medium-sized cities. The scenario of affected farmers at lower levels (wereda towns and kebele centers) is not assessed. Thus, further research is required to see the status quo of land expropriation and compensation practices in these areas. Besides, the existing reality of expropriation and compensation payments, the pitfalls of practicing the expropriation rules, and the drawbacks of the expropriation and compensation legislation require sound and smart expropriation legislations which should be updated from time to time based on relevant and reliable data which can be supported by information and communication technology (ICT). Further research is needed on this topic to meet the challenges of expropriation in particular, and to contribute to Smart Cities and Smart Villages Research in general, as noted by Visvizi and Lytras [43]. Nevertheless, this study contributes to the literature about land expropriation and compensation in developing nations, and it can be used as an input for policy makers to develop smart expropriation and compensation legislation that considers the life improvement strategies for affected peri-urban households.

**Author Contributions:** Conceptualization, S.K.A. and R.M.; Data curation, S.K.A.; Formal analysis, S.K.A. and R.M.; Investigation, S.K.A.; Methodology, S.K.A.; Project administration, R.M.; Resources, R.M.; Writing—original draft, S.K.A.; Graphics: R.M.; Writing—review & editing, R.M.; Funding acquisition: R.M and S.K.A. All authors have read and agreed to the published version of the manuscript.

**Funding:** This research was funded by the Austrian Development Cooperation within the Austrian Partnership Program in Higher Education and Research for Development (APPEAR). Project no. 113 "Implementation of Academic Land Administration Education in Ethiopia for Supporting Sustainable Development" (EduLAND2).

**Conflicts of Interest:** The authors declare no conflict of interest.

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
