# Peer review of "Community Involvement and Compensation Money Utilization in Ethiopia: Case Studies from Bahir Dar and Debre Markos Peri-Urban Areas"

_sustainability, doi:10.3390/su12114794_

Round 1

Reviewer 1 Report

The authors present an important problem of imperfect rules and practices of expropriation of inhabitants of selected peri-urban zones in Ethiopia.

It is surprising, however, that the authors do not refer to quite a number of publications related to their subject and country, and sometimes precisely to their place of research (Bahir Dar). I include examples of 10 publications at the end of the review suggesting that they should be used in Literature review and/or Discussion. It should also be very clearly emphasised when presenting the aims of the work, how this study differs from other, analogous studies, e.g. in the Bahir Dar region - whether it is not an unnecessary duplication of studies already carried out by other authors. What is the originality, novelty of this research?

The article contains a number of technical shortcomings in relation to the journal's requirements, including: incorrectly marked author for correspondence (judging from the e-mail address), text alignment, citations of literature in the text, construction of tables. The English itself also needs to be improved, many words are used incorrectly (in terms of meaning or grammar) or seem to be completely inconsistent with the intentions of the authors (vice versa). In many cases the sentence structure is not correct, and some spaces are missing.

Specific comments on the text (if I ask questions below, I hope they will be answered in the text of the article and not just in response to the review, because I ask these questions on behalf of the readers)

Abstract:

L19-20 – why „the necessary technical and administrative support is detrimental (= bad!) for the proper utilization of the compensation money” ?

Introduction:

L27 (and further on in the text) – populace? probably rather population

L30-31 - The urban administration is expropriating peri-urban agricultural lands payment of compensation. – how can payment be expropriated? you need to improve the sentence structure

L31-32 – „This study is designed to assess the adequacy of the compensation payment”. It should be clearly added that this is a subjective assessment of the persons surveyed and not made separately by the authors of the article, based on their own evaluation of the value of the land

L36-38 – „The basic principle of payment of adequate compensation is to keep the landholders in the same economic position after the land is expropriated (Erasmus, 1990; FIG, 2010).” – in what timeframe? for the rest of their lives?

L41-42 – why reasonable compensation is detrimental?

Literature review:

L81-83 – lack of information sources

L109 – why is Protection a capital letter?

L138 – rather „In Ethiopia”, than „In Ethiopian”?

L145 – please explain the term „the Derg era”

L157 – please explain the term „wereda”

Study areas and methodology

L180 – unify the spelling of DebreMarkos (the title of the article is Debre Markos)

L180-184 – this paragraph is not very consistent. If the area under study is within 5 km of a built-up area, why are there also built-up (residential) areas? In such a case, this 5 km zone should be counted from them. Maybe it is a 5 km radius from the border of dense urban areas?

L186 – no explanation of the abbreviation ANRS

L192 – „KidaneMihret” or „Kidane Mehret”?

L218-223 – how were the respondents selected? was it a group of all those who were expropriated? and if not, on what basis were they selected for research and what proportion of all expropriated persons might it be? How many people were interviewed in a given region? What questions were they asked?

L224-232 – how many people were there in total and in the individual examined regions? how specifically were they selected? what questions were asked?

When were all these surveys and interviews conducted? There is also a lack of information on how the results were analysed. In principle, the methodology should be described in more detail.

Results and discussion

L239-240 „But the current main segment of expropriation in the study peri-urban areas is in order to cater land required for urban expansion.” – is this something very different from the „public roads, parks, schools and health care” mentioned above? Please specify this

L259-261 „For instance, Haregeweyn et al. (2012) have noted that from their Bahir Dar peri-urban study households, on average every respondent has lost 0.89 hectare of land” – how did they get this data - asking the same people as you? so can it be said that the expropriation of land from a person is graduaÅ‚ (from 0.89 to 0.96)? unless it is a one-off decision - how does it work in practice? Has the city boundary, from which the radius of the area studied by the authors of this publication was determined, moved since their (Haregeweyn et al. ) research? The answer to these questions may help to determine whether there are any trends, regularities in the size of expropriations during these 8 (?) years.

L302-307 – were they people who were interviewed (experts described in lines 224-232)? however, the Methodology mentions the questionnaires sent out, and here it refers to an interview, which would indicate that this is a different group of people. It would be good to explain this!

L314-316 – what exactly is the participation of entire communities in land expropriation and compensation? Because later in the text there is a statement: „Instead of the involvement of the community, local representatives represented the society during expropriation and compensation processes.”. So what is the difference between one solution and the other, why is the latter worse (as the context suggests)?

L329 – please explain the term „kebele office”

L330-332 – here, as in the case of Haregeweyn et al. (2012), it would be worthwhile to analyse more closely the similarities, differences, dependencies and trends between the authors' own results and Alemu's results for the Bahir Dar

L337-339 – how much time must landholders be informed about the time of expropriation in advance? How much time do they have to prepare for this event?

L346-347 – „until the issue is decided” – how is this decided? does the landholder have a chance to get compensation that will satisfy him?

L352-354 – again the same case of similar research from the Bahir Dar region without further analysis

L378 and L407 – this information should be in the Methodology

Subchapter 4.3 – It is a regrettable fact that the authors have not attempted, at least in a few examples (parcels), to assess the real value of the expropriated land in order to be able to relate to this feeling of the owners of that land - do they really have reason to believe that they have been affected.

Table 2 – the last cell of the Table should probably contain 100.0% rather than 80.8%

L422 – table 1, not table 9

L427-428 – It would be worthwhile to detail this information from 2012. (besides, there is an error in the name)

L438-440 – does this information come from the authors' research or from the cited publication about Addis Ababa? because it is unclear

L450-451 – „But there is no extra land to be used for land to land compensation in their locality.” – which locality?

L452-453 – why was the solution given by some farmers funny?

L459 – to organize what?

L481 – which respondents? from Kasa’s publication?

The titles of subsections 4.4 and 4.5 should be differentiated - it should be added that the first is from the perspective of landholders, and the second from the perspective of experts

L504-513 – it fits the Methodology better

Conclusions

L582-584 – lack of source of information

It is also worthwhile to review the following items and use them in the Literature review and/or Discussion:

  • The History of Expropriation in Ethiopian Law DOI: 4314/mlr.v7i2.4
  • Take out the farmer: An economic assessment of land expropriation for urban expansion in Bahir Dar, Northwest Ethiopia https://doi.org/10.1016/j.landusepol.2019.104038
  • Compensation Practices in the Ethiopian Expropriation Process http://lup.lub.lu.se/luur/download?func=downloadFile&recordOId=8895396&fileOId=8895399
  • Land Valuation for Expropriation in Ethiopia: Valuation Methods and Adequacy of Compensation https://chilot.me/wp-content/uploads/2011/04/land-valuation-for-expropriation-in-ethiopia1.pdf
  • Expropriation, valuation and compensation practice in Ethiopia: The case of Bahir Dar city and surrounding https://doi.org/10.1108/02637471311309436
  • Public Purpose as a Justification for Expropriation of Rural Land Rights in Ethiopia DOI: https://doi.org/10.1017/S0021855315000285
  • Land Expropriation and Compensation Payment in Ethiopia: Review https://www.iiste.org/Journals/index.php/JEDS/article/view/24279
  • Compensation for Expropriation in Ethiopia and the UK: A Comparative Analysis http://www.fig.net/resources/proceedings/fig_proceedings/fig2014/papers/ts01f/TS01F_ambaye_6821.pdf
  • Urbanization in Ethiopia: Expropriation Process and Rehabilitation Mechanism of Evicted Peri-Urban Farmers (Policies and Practices) https://www.hilarispublisher.com/open-access/urbanization-in-ethiopia-expropriation-process-and-rehabilitation-mechanism-of-evicted-periurban-farmers-policies-and-practices-2162-6359-1000451.pdf
  • REAL PROPERTY VALUATION IN EXPROPRIATION IN ETHIOPIA: BASES, APPROACHES AND PROCEDURES https://revues.imist.ma/index.php?journal=AJLP-GS&page=article&op=view&path%5B%5D=14233

Author Response

Thank you Dear Associate Editor and all reviewers for your valuable contributions to enrich the paper. We tried to address all issues in an effort to improve the paper. Replies to your comments are presented below.

“Assessment of Community Involvement and Compensation Money Utilization in Ethiopia: Case Studies from Bahir Dar and Debre Markos Peri-Urban Areas”

Thank you Dear Associate Editor and all reviewers for your valuable contributions to enrich the paper. We tried to address all issues in an effort to improve the paper. Replies to each review comment are presented below.

Reviewer 1

The authors present an important problem of imperfect rules and practices of expropriation of inhabitants of selected peri-urban zones in Ethiopia. It is surprising, however, that the authors do not refer to quite a number of publications related to their subject and country, and sometimes precisely to their place of research (Bahir Dar). I include examples of 10 publications at the end of the review suggesting that they should be used in Literature review and/or Discussion. It should also be very clearly emphasised when presenting the aims of the work, how this study differs from other, analogous studies, e.g. in the Bahir Dar region - whether it is not an unnecessary duplication of studies already carried out by other authors. What is the originality, novelty of this research?

Dear Reviewer! Thanks for providing relevant literatures for reference. We included most of them. However, it should not be surprising to include all references related to expropriation and compensation in one small article with word count limits. There are some studies conducted in expropriation and compensation especially in peri-urban Bahir Dar. Most of them are on the amount of compensation. This research paper has taken peri-urban Debre Markos (medium size city) besides Bahir Dar (relatively big city). But we don’t think that these are enough. Even forwards, other similar research papers shall be conducted including wereda towns and kebele centers. For instance, in Bahir Dar, even if dissatisfaction on the amount, all have got compensation but in medium sized Debre Markos, there are some households who relinquished their parcels without getting even that small amount of compensation.  Because, these are valuable inputs for policy makers to design expropriation and compensation legislation fit to purpose. However, the main focus of our research is on the support provided to effectively use the compensation money for income generating businesses and involvement of the affected households in the process. Up to our understanding, it is original research paper which investigated the current expropriation and compensation praxis, and recommended solutions for the shortcomings identified during the research.

The article contains a number of technical shortcomings in relation to the journal's requirements, including: incorrectly marked author for correspondence (judging from the e-mail address), text alignment, citations of literature in the text, construction of tables. The English itself also needs to be improved, many words are used incorrectly (in terms of meaning or grammar) or seem to be completely inconsistent with the intentions of the authors (vice versa). In many cases the sentence structure is not correct, and some spaces are missing.

Dear reviewer now we have revised the article according to the requirement of the journal. The correspondence author is revised as noted; text alignment, citations, tables are revised according to the journal requirement. We have also tried to revise the language critically reviewing the content of the article. The sentence structure is revised according to the comments. Concerning the spaces, the unification of some words was created since the article was written using Microsoft word 7). Now, it is edited and we hope the problem will be resolved.

Specific comments on the text (if I ask questions below, I hope they will be answered in the text of the article and not just in response to the review, because I ask these questions on behalf of the readers)

Abstract:

L19-20 – why „the necessary technical and administrative support is detrimental (= bad!) for the proper utilization of the compensation money” ?

Thanks! It was mistakenly used and now revised according to the comment

Introduction:

L27 (and further on in the text) – populace? probably rather population

As far as we know, populace can be used normally as it is used. However, if it created some confusion, it is necessary to edit and substituted with population as proposed.

L30-31 - The urban administration is expropriating peri-urban agricultural lands payment of compensation. – how can payment be expropriated? you need to improve the sentence structure

  • The structure is revised and improved

L31-32 – „This study is designed to assess the adequacy of the compensation payment”. It should be clearly added that this is a subjective assessment of the persons surveyed and not made separately by the authors of the article, based on their own evaluation of the value of the land

  • Now it is revised according to the comment

L36-38 – „The basic principle of payment of adequate compensation is to keep the landholders in the same economic position after the land is expropriated (Erasmus, 1990; FIG, 2010).” – in what timeframe? for the rest of their lives?

  • Dear review, we don’t think it is necessary to put timeframe since it is not known what will happen afterwards. What is expected from the compensation payment is it should be adequately valued and the compensation money should be used to a business that can substitute income what they lost from that parcel.

L41-42 – why reasonable compensation is detrimental?

  • The word detrimental was used mistakenly in the paper and now it is revised.

Literature review:

L81-83 – lack of information sources

  • Dear Reviewer, the source of the information is ‘Wikipedia’. Most of the time, the Wiki sources are taken as a general truth and that is why we didn’t mention the source

L109 – why is Protection a capital letter?

  • Edited with small ‘p’ letter

L138 – rather „In Ethiopia”, than „In Ethiopian”?

  • Revised by correcting the structure of the sentence

L145 – please explain the term „the Derg era”

  • Now, it is explained in a better way.

L157 – please explain the term „wereda”

  • Even if the word ‘wereda’ is available in Wikipedia sources, what it means is explained according to the review comment to avoid confusion of readers

Study areas and methodology

L180 – unify the spelling of DebreMarkos (the title of the article is Debre Markos)

  • Corrected by separating the two words as ‘Debre Markos’

L180-184 – this paragraph is not very consistent. If the area under study is within 5 km of a built-up area, why are there also built-up (residential) areas? In such a case, this 5 km zone should be counted from them. Maybe it is a 5 km radius from the border of dense urban areas?

  • Right there are sparsely located rural houses and accordingly to revise with your comments ‘dense urban areas’ seems sound and we revised it accordingly.

L186 – no explanation of the abbreviation ANRS

  • is explained as Amhara National Regional State (ANRS)

L192 – „KidaneMihret” or „Kidane Mehret”?

  • As stated above, the unification problem of some words was due to using old version of Microsoft word. Now, by this chance, I revised not only the word to ‘Kidane Mihret but also the Microsoft word is upgraded to window 10.

L218-223 – how were the respondents selected? was it a group of all those who were expropriated? and if not, on what basis were they selected for research and what proportion of all expropriated persons might it be? How many people were interviewed in a given region? What questions were they asked?

  • Information about the respondents’ data is incorporated in the methodology part of the article according the requests. Please be informed that this data is part of the lengthy dissertation.

“In order to select the study households, initially, the total household list of the study peri-urban areas was obtained and recorded from the concerned land administration offices and agricultural development agents’ offices. From the list, 2386 households that lost land through expropriation are identified and recorded with the support of kebele (smallest administrative unit) land administration committees. Respondents were selected randomly from the expropriated household lists in both study areas. In total, 269 respondents were selected and interviewed with the aim to assess the expropriation and compensation practices. Out of these, 101 are from Debre Markos peri-urban areas whereas the remaining 168 from Bahir Dar peri-urban areas.”

Dear reviewer, pardon please! We don’t believe that it is necessary to include in the article such a lot of questions since it adds no value to the article.

L224-232 – how many people were there in total and in the individual examined regions? how specifically were they selected? what questions were asked?

  • The population number is noted in the above question. But a lot of questions were prepared for the data collection. All those and other questions for the other research questions of the dissertation are available at the back of the dissertation. We don’t think, the research questions are necessary for this article since the word content becomes large. If these research questions are deemed necessary for this article, it is possible to attach as a supplement of the paper at the end. But, we don’t believe on its necessity.

When were all these surveys and interviews conducted? There is also a lack of information on how the results were analysed. In principle, the methodology should be described in more detail.

  • The data collection is done in 2014. As described earlier, this article is part of the dissertation and it was collected during that time. So, the data presented here is from 2000 up to 2014. The data collection period is described in the article. This article gives information what the expropriation and compensation practice during that period of time looks like and also helps to analyze what the situation looks like afterwards. The methodology has been described in more detail according to the valuable comments of the reviewers.

Results and discussion

L239-240 „But the current main segment of expropriation in the study peri-urban areas is in order to cater land required for urban expansion.” – is this something very different from the „public roads, parks, schools and health care” mentioned above? Please specify this

  • It is to state that most developed countries expropriate land when it is required for public facilities, infrastructures… and the like. But, in Ethiopia, since the government is the sole proprietor of land and also the rate of spatial urban expansion is high and accordingly the expropriation is for buildings of different uses (residential main segment, commercial, industrial). It does not mean that utilities and infrastructures associated with this are not built. For instance, if railway is planned to be constructed in Bahir Dar, it will create many distractions to the existing buildings but assuming the purpose the railway contributes to the community, government takes expropriation measure on real properties. It is to mean that and now some explanations are included in the article.

L259-261 „For instance, Haregeweyn et al. (2012) have noted that from their Bahir Dar peri-urban study households, on average every respondent has lost 0.89 hectare of land” – how did they get this data - asking the same people as you? so can it be said that the expropriation of land from a person is graduaÅ‚ (from 0.89 to 0.96)? unless it is a one-off decision - how does it work in practice? Has the city boundary, from which the radius of the area studied by the authors of this publication was determined, moved since their (Haregeweyn et al. ) research? The answer to these questions may help to determine whether there are any trends, regularities in the size of expropriations during these 8 (?) years.

  • It is not assumed that we will ask the same people at different times. When you judge from the reality, if we don’t work at the same time consulting each other that we could contact the same persons, how it could be? Even we do no authors facially but we got their published in journals. Both of us take sample from the expropriated households. Even though there might be probability for some respondents to be part of the study, the majority will be different respondents. Dear Review, please be free about your hesitation about the two independent studies. Anyways, the reviewer is interested to see the trend of peri-urban land tenure transformation. It is briefly included in the text part that the trend shows increasing. Even though the article is sent to a journal, it is not long period since the data collected. Probably 3 to 4 years difference. During this period of time the trend shows increasing.

L302-307 – were they people who were interviewed (experts described in lines 224-232)? however, the Methodology mentions the questionnaires sent out, and here it refers to an interview, which would indicate that this is a different group of people. It would be good to explain this!

  • It is explained and put in the methodology part. To give some information, these are experts in both study areas. Some are from municipalities and others are land administration office in the urban administration. Anyways, it is described in the methodology part.

L314-316 – what exactly is the participation of entire communities in land expropriation and compensation? Because later in the text there is a statement: „Instead of the involvement of the community, local representatives represented the society during expropriation and compensation processes.”. So what is the difference between one solution and the other, why is the latter worse (as the context suggests)?

  • This part is also revised in the article. However, the local administrative bodies should not represent the affected farmers in expropriation and valuation processes since those affected ones need to be discussed and consulted for the necessary information instead of kebele administrative bodies. The participation of the kebele administrative bodies is not fault but their act on behalf of the expropriated ones is mistake.

L329 – please explain the term „kebele office”

  • Explained as commented

L330-332 – here, as in the case of Haregeweyn et al. (2012), it would be worthwhile to analyse more closely the similarities, differences, dependencies and trends between the authors' own results and Alemu's results for the Bahir Dar

We have tried to show the trend that it is not improving. Even if possible to see in the article, for the easy information of the reviewer, we put below what incorporated in the article.

“Other studies also documented similar problems with respect of the involvement of the affected community [34, 35, 14, 30]. This study result and other similar findings indicate that even though participatory approaches are necessary in expropriating of land for public purposes, the reality at the ground indicates that it is overlooked by the municipalities. The trend should indicate improvement but as has been noted in the current studies too, significant improvement is not observed.”

L337-339 – how much time must landholders be informed about the time of expropriation in advance? How much time do they have to prepare for this event?

  • It is before three months and it is included in the article

L346-347 – „until the issue is decided” – how is this decided? does the landholder have a chance to get compensation that will satisfy him?

  • How the issue to be decided is incorporated in the article as follows.

“If the landholder refused to take the money assuming that it is not properly valued, he can present his complaint to the grievance hearing administrative organ of the urban administration or to the court office in rural areas [26]. If still dissatisfied with the decision, he can appeal to the appeal court. If the size of the parcel is not properly considered and the crops value not properly estimated the decision making bodies have a right to make adjustments by assigning other experts who will scrutinize the case. Even if the generally, the money is low, affected people get better feeling when their grievance is handled properly.”

L352-354 – again the same case of similar research from the Bahir Dar region without further analysis

  • The analysis is already done.

L378 and L407 – this information should be in the Methodology

  • We don’t think this should be in the methodology. It is the result of the study on the timeliness and amount of compensation

Subchapter 4.3 – It is a regrettable fact that the authors have not attempted, at least in a few examples (parcels), to assess the real value of the expropriated land in order to be able to relate to this feeling of the owners of that land - do they really have reason to believe that they have been affected.

  • It is known that the lease prices in these areas are very high compared to the compensation payments. But the study of this is to see the perception of farmers and experts about the current amount of compensation payments. Anyways, even though it requires independent study to see the disparity between lease prices and compensation payments, However, in general terms the following is included in the discussion:

“In Ethiopia land is not a free commodity subject to exchange through sales. Because of this it is difficult to explicitly compare the price of land and the compensation payments. But there are illegal land transactions in the peri-urban areas. In addition, currently the transfer of urban land for different uses has been by means of leases. So, it is possible to make rough comparison between the compensation payments with the price of the land in illegal transactions and lease prices to judge the compensation payments. From what is practically observed the prices for illegal transactions and lease is very high. The farmers’ dissatisfaction increases when they hear the high lease prices of the adjacent parcels. Accordingly, they hate expropriation to much extent try to get advantages even by transacting their land illegally. Illegal transaction is not secure especially for the buyer since if the seller denies the transaction, the buyer loses his money. It is based on trust and religious taboos what they did this transaction. Even under such uncertainties, the illegal sales price is by far better than the compensation payments and accordingly, peri-urban farmers are motivated to undertake illegal sales before their land becomes expropriated..”

Table 2 – the last cell of the Table should probably contain 100.0% rather than 80.8%

  • It is not the total one. It is response for “Have you got the compensation money in time?”

L422 – table 1, not table 9

  • Corrected

L427-428 – It would be worthwhile to detail this information from 2012. (besides, there is an error in the name)

  • Modified according to the comment

L438-440 – does this information come from the authors' research or from the cited publication about Addis Ababa? because it is unclear

  • It is cited from the authors and the references are put

L450-451 – „But there is no extra land to be used for land to land compensation in their locality.” – which locality?

  • ‘Locality’ is the village where they are living. So, substituted by village

L452-453 – why was the solution given by some farmers funny?

  • Now it is updated in a better way. For us it was impressive when farmers generate such kind of noble idea of being engaged in mechanized farming. Most of the time, they request the government to give them land through from adjacent areas through land redistribution. There is also shortage of arable land for such kind of requests. But they are interested to cultivate unoccupied land in low land areas and this was impressive for us.

L459 – to organize what?

  • Corrected with “organize in groups”

L481 – which respondents? from Kasa’s publication?

  • Yes, right! It was a bit confusing. Now, it is revised in a better way. The respondents are not kasa’s. They are the respondents of this study.

The titles of subsections 4.4 and 4.5 should be differentiated - it should be added that the first is from the perspective of landholders, and the second from the perspective of experts

  • Now, both the titles are corrected

L504-513 – it fits the Methodology better

  • Shifted to methodology part

Conclusions

L582-584 – lack of source of information

  • Now, the result is presented in table and the information became full.

Reviewer 2 Report

Dear Authors,

in reviewers opinion several technical issues needs to be improved:

  • Improve citations according to magazine requirements
  • From versus 138, expropriation has been reported in Ethiopia. I think this should be further expanded. It could also be described in a separate chapter (but it is up to authors). Is the expropriation  allowed by the Ethiopian constitutions? If so, then you should start descriptively from higher-level acts and move on to lower-level acts. In the text, information about expropriation is found in various places and it is difficult for the reader to understand the procedure of expropriation in Ethiopia (that is why i suggest make expropriation as a separate point)
  • I propose in the Study Areas to add a map with the marked places where the research was conducted (if not - that is not something that diminishes the material)
  • Is ANRS an abbreviation??
  • Is there no norm in Ethiopia under which an expropriated person may recover the expropriated property if the land was not used for expropriation.
  • Can the authors not propose a solution whereby the expropriated magician use the expropriated land until the investment is started on it?
  • in verse 422, table 9 is mentioned which is not in the article
  • please improve the tables

I think that the following information should be added to the article:

  • what's innovative about the article;
  • what the authors suggest, some solutions;

At the moment, this is a description of the phenomenon without proposals for changes or improvements. Maybe you should think about the method of assessing changes?

Author Response

Thank you Dear Associate Editor and all reviewers for your valuable contributions to enrich the paper. We tried to address all issues in an effort to improve the paper. Replies to each review comment are presented below.

Reviewer 3 Report

The manuscript entitled “Assessment of Community Involvement and Compensation Money Utilization in Ethiopia: Case Studies from Bahir Dar and Debre Markos Peri-Urban Areas”, by S.K. Agegnehu and R. Mansberger, presents an interesting work.

 It needs some significant improvement. Some suggestions are as follows:

  1. Please follow the journal author instructions. It would be useful for the reader to follow the classical text structure (i.e. Introduction-methodology-results-discussion-conclusions. A better presentation of your results and an extensive discussion would improve your paper.
  2. Please use different terms in the “Title” and the “Keywords”.
  3. Please do not use references in the “Conclusions” section.
  4. The “Conclusions” section needs rewriting.
  5. It would be useful to be described the aim of this paper.
  6. The English language usage should be checked by a fluent English speaker. It is suggested to the authors to take the assistance of someone with English as mother tongue.
  7. You could insert some maps.
  8. I propose to the authors to be more specific, explanatory and simplified in order to be easily understandable from the readers.
  9. You could enrich the scientific literature.
  10. Please justify convincingly why this manuscript (method, thematology etc) connected with Sustainability’s content and scope. Perhaps the using of proper literature from this jou would be helpful.
  11. I suggest to the authors to use more international references.
  12. The authors could make a discussion about the relationship between urban planning and hazard assessment using the following publications:

- Bathrellos, G.D., Skilodimou, H.D., Chousianitis, K., Youssef, A.M., Pradhan, B. (2017): Suitability estimation for urban development using multi-hazard assessment map. Sci Total Environ, 575: 119 – 134.

- Bathrellos, G.D., Gaki-Papanastassiou, K., Skilodimou, H.D., Papanastassiou, D., Chousianitis, K.G. (2012): Potential suitability for urban planning and industry development by using natural hazard maps and geological - geomorphological parameters. Environmental Earth Sciences (Environ Earth Sci), 66 (2): 537 – 548.

13. Correct references in the text and the reference list according to the journal’s format. Please format the references’ list by using the correct journal abbreviations.

See the following link: https://images.webofknowledge.com/images/help/WOS/A_abrvjt.html

14. Please be careful with the spaces between the words.

Author Response

Thank you Dear Associate Editor and all reviewers for your valuable contributions to enrich the paper. We tried to address all issues in an effort to improve the paper. Replies to your review comments enclosed.

Reviewer 4 Report

I think the authors address an interesting topic for Ethiopia and other developing countries.

I recommend to include maps of the studied areas with geographic references to the country and continental area.

I have noticed that references are in alphabetical order. I am not sure if this is the style required by the journal.

Some bibliographical references could be useful in line with local agreements and institutional approaches. I recommend, for instance (among others), Vázquez-Barquero and Rodríguez-Cohard (2016): Endogenous development and institutions: Challenges for local development initiatives, Environment and Planning C: Government and Policy, 34(6):1135-1153, in order to discuss in the conclusion section.

My main concern is about the organisation of the paper. I recommend to separate results and discussion in order to better understand the contribution of authors and their recommendations based in the literature review and other cases published. Even some paragraphs in Results and Discussion section would be better located in the introduction section, such as since line 235 to 253, for instance. From line 505 to 513 I think this paragraph would go better to Method section.

Minor errors: I recommend to explain the acronym ANRS; in line 227 say many instead of money; in line 422 say table 9 instead of table 1; in line 452 and 453 it is strange to say that the solution provided for farmer is amusing.

Author Response

Thank you Dear Associate Editor and all reviewers for your valuable contributions to enrich the paper. We tried to address all issues in an effort to improve the paper. Replies to your review comments are enclosed.

“Assessment of Community Involvement and Compensation Money Utilization in Ethiopia: Case Studies from Bahir Dar and Debre Markos Peri-Urban Areas”

Thank you Dear Associate Editor and all reviewers for your valuable contributions to enrich the paper. We tried to address all issues in an effort to improve the paper. Replies to each review comment are presented below.

Review 4

I recommend to include maps of the studied areas with geographic references to the country and continental area.

Thanks dear Reviewer! The map is included according to the recommendation

I have noticed that references are in alphabetical order. I am not sure if this is the style required by the journal.

Now, it is arranged according to the requirement of the journal

Some bibliographical references could be useful in line with local agreements and institutional approaches. I recommend, for instance (among others), Vázquez-Barquero and Rodríguez-Cohard (2016): Endogenous development and institutions: Challenges for local development initiatives, Environment and Planning C: Government and Policy, 34(6):1135-1153, in order to discuss in the conclusion section.

Dear reviewer, some more references are included in the revised article. It is an interesting article and we agree with you about the importance of local development policy as an instrument for regional development. Nevertheless, the discussion on this topic would go beyond the scope of the paper.

My main concern is about the organisation of the paper. I recommend to separate results and discussion in order to better understand the contribution of authors and their recommendations based in the literature review and other cases published. Even some paragraphs in Results and Discussion section would be better located in the introduction section, such as since line 235 to 253, for instance. From line 505 to 513 I think this paragraph would go better to Method section.

We have separated the result and discussion parts according to the comment. According to the comment some paragraphs from the discussion section transferred to the introduction section. For instance, “Community participation in land expropriation and compensation builds trust and reduces land disputes [23]. Accordingly, communities affected with expropriation should effectively participate in the processes of expropriation and compensation [5]. It is explicitly stated in Proclamation 455/2005 that it is mandatory to inform the landholders with written letter about the time of expropriation and the amount of compensation payment.”

Besides, those to be shifted to the methodology section are done accordingly. For instance, “The survey questions were delivered to 26 professionals. Out of those, 3 did not return the questionnaire in time. All survey employees are from municipalities and newly established rural land administration processes in the urban administration. All experts have exposures to the expropriation and compensation practices and rules with different years of experience in the institution, the minimum being 6 months and the maximum 7 years. The survey employees list comprises both experts and coordinators (vice heads, process owners) at various levels. About 22% of them are coordinators of land administration processes at different levels. Briefly, the survey group is representing professionals at management and technical level with various levels of experience. “  this part is shifted to the methodology part.

Minor errors: I recommend to explain the acronym ANRS; in line 227 say many instead of money; in line 422 say table 9 instead of table 1; in line 452 and 453 it is strange to say that the solution provided for farmer is amusing

  • ANRS
  • is explained as Amhara National Regional State (ANRS)
  • In line 227 say many instead of money
  • Revised as noted

  • in line 422 say table 9 instead of table 1
  • revised as noted
  • in line 452 and 453 it is strange to say that the solution provided for farmer is amusing
  • The kind of solution proposed by subsistence farmers was surprising. That is why we said ‘the solution provided by farmers was amusing’. Anyways, if it creates confusion we believed its revision. Accordingly, ‘amusing’ is substituted by ‘impressive’.

Reviewer 5 Report

The article deals with an interesting topic. The results are interesting too.

However, the following issues should be taken up in the revision:

  • The novelty of the article and the results are not obvious. They have to be highlighted. It has to be explained where the article goes beyond the state of knowledge and the quoted literature.
  • The article describes a lot of interesting facts regarding expropriation and compensation in the two cities. However, the authors do not clearly formulate objectives and research qwuestion which the article deals with. Moreobver, its theoretical embeddedness is not made clear (lines 69 ff.). Main hypotheses, which the authors want to address, have to be described and derived from literature. Its contribution to theory should be made clear, even if t is mainly empirical.
  • The literature review (lines 69-177) mainly deals with land management issues in general and in Ethiopia. However, the state of the art regarding the utilization of compensation money is not dealt with in an appropriate way.
  • The case study areas (lines 179 ff) are described but it is not made clear why they were chosen, what we can learn from the results which were reached there, and whether the results can be considered as representative.
  • A map regarding the location of the two cities and available land for compensation (lines 582-584) would be helpful for the reader.
  • It is not clear what role the climatic indicators in the description of one of the case study areas play (lines 189-191).
  • In the description of the methodology (lines 208 ff), detailed information (e.g., about data sources, number of interviewees, stakehlder groups, interview guidelines) are missing. This chapter should contain the whole methodological information which is partly only mentioned later (e.g., lines 504-513).
  • The results (lines 233 ff.) provide a lot of interesting details. However, it should be made clear which research questions they address, and what we can learn from the results.
  • Community participation (lines 313 ff.) is taken in a rather unreflected way here (lines 314-315). It should be made clear that participation has to be well managed in order to be successful. It is not successful per se.
  • Differences between the two cities are interesting (Table 1, line 377). However, it is not clear what the reader can learn from these differences. There is also no hypothesis regarding this issue.
  • It is not clearly explained how the results are validated. For example, Table 2 (line 406) shows that the vast majority of respondents argue that the compensation payment is "low". But it is not clear what this means. And it is not compared with any objective data in order to be able to assess the perception. Also other "locational differences" (line 426) are not further explained.
  • Table 9 (mentioned in line 422) is not existing.
  • The authors are encouraged to include two more tables regarding the results of chapters 4.4 and 4.5. 
  • For the reader it is not understandable why farmer's solutions are "amusing" (line 453).
  • The conclusions (lines 533 ff) should include the novelty of the results and their contribution to theory and the state of the art. Moreover, the various recommendations (lines 545 ff) which the authors provide should be more clearly structured, highlighted and presented. 

Author Response

Thank you Dear Associate Editor and all reviewers for your valuable contributions to enrich the paper. We tried to address all issues in an effort to improve the paper. Replies to your review comments are enclosed.

Round 2

Reviewer 1 Report

The structure of the article has been improved, a number of necessary explanations have been added and errors have been corrected. Below I have given some more comments and suggested corrections. Moreover, I still think that more of the articles on expropriation in Ethiopia I gave in my previous review can be used.

L89-93 - Wikipedia (according to the authors' commentary) is probably not a very appropriate source of information in a scientific article...

Additionally, directing a reader to Wikipedia to check the meaning of the words in the article is also not entirely correct (response to authors' comments).

L158 – rather „In Ethiopia”, than „In Ethiopian”

L192 – „should be participate”? or „should participate”?

L355 – the title of subsection 4.6 is not well formulated - it should refer to the content and not to the group of respondents

PodrozdziaÅ‚ 4.6 – „This was done to assess the perception of the experts on expropriation and compensation practices and to understand their opinion on the legislation.” Further in the text of the subchapter there is only a reference to the assessment of the current compensation payment, the other results are missing.

L364 – rather Discussion, than Discussions

L386 (and other places in the text) – I think it would sound better: „For instance, Nigusie [34] gave evidence…”, L390 „Kasa et al. [13] also have observed…” etc.

L398 – worth adding at the beginning of the sentence „In this study” lub „In our study”

L410 – in Ethiopian or in Ethiopia?

L553 – to organize (whom - them?) in groups – it's still not entirely clear

L581 – not very well (too briefly) formulated beginning of sentence: „These results support [4] that farmers affected by expropriation will use the money quickly and unwisely…”.

L583-587 – an incomprehensible sequence of words - I guess that should be separated into two sentences? „Studies conducted in peri-urban areas of Addis Ababa also have observed that expropriated farmers have not got support in compensation money utilization either from the governmental authorities or from NGOs [33, 34, 13] also have noted that expropriated farmers did not utilize the compensation money in livelihood improving manner because of lack of “parallel business and skill development intervention”. Moreover, the studies cannot observe because they are not living creatures or devices.

L587 – no closing quotation mark

Instead of making a separate Subchapter 5.6, it would be better to incorporate its contents into the other subchapters of the discussion, so that the issue under discussion is presented comprehensively - both from the perspective of farmers and experts (+ literature).

L673-678 Why is the last name MANSBERGER written in capital letters?

Author Response

Thank you once again for taking time and for your valuable contributions to enrich the paper. We are thankful also for your comments in the second round. We tried to address the outlined issues as far as possible. Nevertheless, we ask for your understanding that we did not consider your suggestions completely. Replies to your review attached. 

Reviewer 3 Report

The manuscript entitled “Assessment of Community Involvement and Compensation Money Utilization in Ethiopia: Case Studies from Bahir Dar and Debre Markos Peri-Urban Areas”, by S.K. Agegnehu and R. Mansberger, presents an interesting and improved work.

The manuscript should be acceptable for publication but some problems could be repaired prior to publication. Some suggestions are as follows:

  1. Please shorten the “Title”. You could transfer the “case studies Bahir Dar and Debre Markos” to the keywords”.
  1. Please do not use references in the “Conclusions” section.
  2. The “Conclusions” section needs rewriting.
  3. It would be useful to be described the aim of this paper.
  4. You could enrich the scientific literature.
  5. Please justify convincingly why this manuscript (method, thematology etc) connected with Sustainability’s content and scope. The using of proper literature from this journal would be helpful.
  6. You could use more international references and not so many theses.
  7. The authors could make a discussion about the relationship between urban planning and hazard assessment using the following publications:

- Bathrellos, G.D., Skilodimou, H.D., Chousianitis, K., Youssef, A.M., Pradhan, B. (2017): Suitability estimation for urban development using multi-hazard assessment map. Sci Total Environ, 575: 119 – 134.

- Bathrellos, G.D., Gaki-Papanastassiou, K., Skilodimou, H.D., Papanastassiou, D., Chousianitis, K.G. (2012): Potential suitability for urban planning and industry development by using natural hazard maps and geological - geomorphological parameters. Environmental Earth Sciences (Environ Earth Sci), 66 (2): 537 – 548.

Author Response

Thank you once again for taking time and for your valuable contributions to enrich the paper. We are thankful also for your comments in the second round. We tried to address the outlined issues as far as possible. Nevertheless, we ask for your understanding that we did not consider your suggestions completely. Please see the enclosed for the responses. 

Reviewer 4 Report

Thanks.

Author Response

Thank you once again for taking time and for your valuable contributions to enrich the paper. We are thankful also for your comments in the second round. We tried to address the outlined issue as far as possible. Please see the attachment for the response. 

Reviewer 5 Report

The quality and clarity of the article has considerally improved.

The explanations and change made by the authors are satisficing.

It has become clearer with which intention the two cases were selected.

It is still questionable whether the results are valid for Ethiopia or just for the Amhara Regional State. This should be clarified and eventually reflected in the title.

Still, the information about the temperature in Bahir Dar should be deleted. It has no direct relation with the topic under discussion. The explanation of the authors is not convincing.

English language and style should be checked by a native speaker.

Author Response

Thank you once again for taking time and for your valuable contributions to enrich the paper. We are thankful also for your comments in the second round. We tried to address the outlined issues as far as possible. Please see the attachment for the responses. 
